# Agreement Analysis between Vive and Vicon Systems to Monitor Lumbar Postural Changes

**DOI:** 10.3390/s19173632

**Published:** 2019-08-21

**Authors:** Susanne M. van der Veen, Martine Bordeleau, Peter E. Pidcoe, Christopher R. France, James S. Thomas

**Affiliations:** 1Department of Physical TherapyVirginia Commonwealth University, MCV Campus, Richmond, VA 23220, USA; 2Department of Neuroscience, Université Laval, Québec, QC G1V 4G2, Canada; 3Department of Psychology, Ohio University, College of Arts & Sciences, Athens, OH 45701, USA

**Keywords:** motion capture, virtual reality, Vive, Vicon, agreement analysis

## Abstract

Immersive virtual reality has recently developed into a readily available system that allows for full-body tracking. Can this affordable system be used for component tracking to advance or replace expensive kinematic systems for motion analysis in the clinic? The aim of this study was to assess the accuracy of position and orientation measures from Vive wireless body trackers when compared to Vicon optoelectronic tracked markers attached to (1) a robot simulating trunk flexion and rotation by repeatedly moving to know locations, and (2) healthy adults playing virtual reality games necessitating significant trunk displacements. The comparison of both systems showed component tracking with Vive trackers is accurate within 0.68 ± 0.32 cm translationally and 1.64 ± 0.18° rotationally when compared with a three-dimensional motion capture system. No significant differences between Vive trackers and Vicon systems were found suggesting the Vive wireless sensors can be used to accurately track joint motion for clinical and research data.

## 1. Introduction

The last decade has witnessed a booming development in immersive virtual reality (VR) technology, where the user has the perception of being physically present in a non-physical environment. These alternative realities can be created by surrounding the user with computer-generated sensory perceptions stimulating vision (e.g., head-mounted displays [1], cave automatic virtual environments [2], etc.), hearing (e.g., virtual acoustics [3], binaural sounds [4], etc.), touch (e.g., haptics technology [5], tactile [6] or force [7] feedback, etc.), smell (e.g., scent cartridges [8,9]), and taste (controlled electrical pulses [10] and aromas [11,12]). 

VR has long been associated with gaming, but now it is expanding into other fields like the healthcare industry. According to a new report by Grand View Research, the VR and augmented reality healthcare market is expected to reach United States Dollars (USD) 5.1 billion by 2025 [13]. VR has been used in numerous biomedical applications from surgical training to medical education, and from psychiatric to motor rehabilitation. Such applications for patient care include the treatment of acute and chronic pain [14,15,16,17,18], specific phobias [19,20], and post-traumatic stress disorder [20,21]. Among several other applications, VR has been successfully used in cognitive and physical rehabilitation after stroke [22,23] and traumatic brain injury [24].

When combined with an optoelectronic three-dimensional (3D) motion capture system, VR becomes a valuable tool to enhance patient motivation during long-term rehabilitation sessions while giving the clinician real-time measurement of joint movement and motor control. However, optoelectronic 3D motion capture systems such as Vicon, can cost up to a hundred thousand US dollars, making this technology out of reach for most clinicians and researchers.

Motion capture systems are used in numerous fields and studies based on motion capture data can be found in biomechanical, sport, and clinical sciences. Optoelectronic motion capture systems based on passive reflective markers were originally developed for gait analysis [25] but this method of motion analysis has also been used to quantify lumbar flexion during reaching tasks [26,27,28]. Optoelectronic motion capture systems provide a powerful measuring tool for biomechanical applications as position accuracy [29] with these systems has reported errors of less than 2 mm [30].

HTC and Valve Corporation released a head-mounted display (HMD) (April 2016), called the HTC Vive. The HTC Vive is an HMD that provides a fully immersive VR environment and costs less than traditional motion tracking systems, such as Vicon. While tracking the orientation and position of the HMD in VR is critical to provide a seamless immersive environment, we found only one recent article that has quantified the precision and accuracy of the position and orientation of the Vive HMD [31] and one evaluating the accuracy of the HTC Vive hand controllers and trackers [32]. Niehorster and colleagues [31] compared the Vive HMD tracking with a Cartesian grid and a WorldViz PPT-X optical tracking system and suggested that tracking accuracy of the HMD display using the Vive was unsuitable for scientific experiments that require accurate visual simulation of self-motion through a virtual world. Likely the larger errors reported were due to tracking of the HMD being lost by the lighthouses (the hardware used to track the HTC Vive HMD and controllers) and tilting of the orientation of the VR space when tracking was re-established. However, others recently found Vive controllers and trackers to have high accuracy (i.e., less than a millimetre) in both static and dynamic planar tasks when additional custom tracking algorithms were applied [32]. Additionally, a recent master’s thesis on the accuracy of Vive tracker position reported a modest error in positional tracking (i.e., 7.5 mm compared to real world positioning and 1cm with respect to a research-grade system) [33]. Although Vive trackers have been shown to have reasonable positional accuracy in static tasks, a more robust examination during dynamic multiplanar tasks is needed to address whether these devices are suitable alternatives for biomedical applications.

As Vive trackers are wireless and lightweight, they can be securely attached to different parts of the body, and therefore it might be possible to accurately track body motion, eliminating the need of an expensive 3D kinematic systems. The aim of this study was to assess the accuracy of position and orientation measures reported by Vive trackers in comparison to an optoelectronic 3D motion capture. The accuracy of the Vive trackers were examined using (1) a robot to produce fixed single plane and multiplanar motions and (2) while attached to the thorax and sacrum of human participants engaging in VR gameplay.

## 2. Materials and Methods

Simultaneous recording of position and orientation of sacral and thoracic marker clusters were made using custom 3D-printed mounting plates that integrated light reflective markers with HTC Vive trackers (see Figure 1b,c). We examined the accuracy of the trackers under two conditions. In the first condition, sacral and thoracic marker clusters were mounted on to a robot arm (SCORBOT) that made single plane and multiplanar movements. In the second condition, the sacral and thoracic marker clusters were attached to healthy participants (n = 7) during VR gameplay.

### 2.1. Motion Capture Systems

#### 2.1.1. Vive System

The Vive kinematic system employed in this experiment consisted of two HTC Vive wireless trackers (HTC America, Inc., Seattle, WA, USA) (Table 1). The position and orientation of these trackers were determined by two fixed infrared laser emitter “Lighthouses” (HTC Vive Lighthouses, Valve Washington, DC, USA). Our experiments used first generation (1.0) “lighthouses” during the human participant game play, and second generation (2.0) for robot simulated movements.

First generation (1.0) “Lighthouses” use infrared lasers that alternately sweep horizontal and vertical lines of light through a 120° field-of-view collection volume. A synchronization pulse at the beginning of each sweep resets the internal clock of trackers to time-zero. The time from the start of a sweep to beam detection by a surface-mounted photodiode infers the angular position of the tracker relative to the fixed “Lighthouse” location. Multiple photodiode detectors on each tracker provide additional orientation information. Time-stamped data is then transformed to a world reference frame and recorded. Second generation (2.0) “Lighthouses” use the same laser projections, but synchronization flashes are eliminated by the use of a synch-on-beam signal to provide angular position information, eliminating the need for a time-based calculation [34].

Using custom software developed within the Unity game engine, (version 2018.2.6f1 Unity Technologies, California, CA, USA), the 6- degrees of freedom (DOF) kinematic data collection from the trackers was set to a sampling frequency of 100 Hz. However, variations in software and hardware processing resulted in actual sampling frequencies ranging 58–100 Hz. Since each sample was time stamped, data were adjusted in post-processing using linear interpolation methods within the Matlab environment (version R2018a Mathworks, Natick, MA) to ensure a 100 Hz output.

#### 2.1.2. Vicon System

The Vicon kinematic system consists of ten Vicon Bonita 10 cameras (Vicon Motion Systems Ltd., Oxford, UK) which track the 6-DOF position and orientation of light-reflective marker clusters mounted on the custom 3D-printed mounting plates (on which the HTC Vive trackers were co-located). These plates were attached to thorax and sacrum of human participants and attached to the robot using elastic straps. The 6-DOF kinematic data from the light reflective markers were collected at a sampling frequency of 100 Hz (with a spatial resolution of 0.1 mm) and recorded using Motion Monitor software (Innovative Sports Training, Chicago, IL, USA) in the Euler Angle sequence y, z, x (Figure 2).

### 2.2. Immersive Virtual Reality Environment

We created a custom virtual reaching game, called *Matchality* (see Appendix A), using Unity software. The game requires players to reach to a static set of cubes, arranged in a 4 × 4 grid, at a self-selected pace. The locations of the cubes in the virtual space are such that the participant could touch the highest row of cubes with 15° of lumbar flexion, the second through fourth rows would necessitate 30°, 45°, and 60° of lumbar flexion, respectively. These target positions are based on a standardized algorithm that takes into account the anthropometrics of the participant [35] but given the number of body segments involved in whole-body reaching, kinematic redundancy allows for successful completion of the task with an infinite number of movement strategies. Game play begins with the random illumination of two cubes for 100 ms each. Participants must then move such that their avatar touches the previously lit cubes in the same sequence as presented. After each successful completion, the pattern is repeated with an additional cube added to create a longer sequence. The sequence length continues to increase until the player is unable to correctly match the sequence, and then the game reverts to a sequence of only two cubes. This is a time-based game that lasts 90 s per set. There are two sets per level and three levels per game. The HTC Vive system (HTC America, Inc., Seattle, WA, USA) was used to allow physical movement of the virtual avatar within the virtual environment. The participants were immersed using a wired HMD (470 g) that uses an organic light-emitting diode display and provides a resolution of 1080 × 1200 per eye, with a refresh rate of 90 Hz, and a field of view of 110°. See Appendix A for an example of the *Matchality* game.

### 2.3. Measurement Set Up

Two HTC Vive lighthouses and ten Vicon 10 cameras were positioned around a 2.5 m by 2.5 m platform providing an adequate data collection volume for unconstrained game play (*Matchality* is a stationary game, when a VR game requires greater player movements, larger virtual areas may need to be covered). The axes of the world reference frame for the Vicon system was such that positive *z* faces upward, positive *x* faces forward, and positive *y* faces leftward relative to the position of the SCORBOT and of the subject. Thus, flexion of the spine would result in a clockwise rotation about the *y*-axis and a forward displacement along the *x*-axis. Twisting of the trunk would result in a rotation about the *z* axis with minimal displacement along the *x*- or *y*-axes.

### 2.4. Robot Instrumentation

To evaluate the accuracy and repeatability of position and orientation measurements in a controlled environment, we used a SCORBOT ER VII (Eshed Robotec/ RoboGroup, Rosh Ha’Ayin, Israel) to rotate the HTC Vive trackers to known positions that require angular rotation about the *y-* and *z*-axes (simulating spine flexion and rotation). Note that rotation about the *y*-axis results in a significant displacement among the *x-*axis while rotation about the *z*-axis has minimal displacement along the *x* and *y*-axes. As illustrated in Figure 3, the SCORBOT ER VII is a 5-axis robot allowing controlled 6-DOF movement of the distal link. The thoracic marker plate was placed on the end-effector (or gripper as labeled in Figure 3) and the lumbar marker plate was placed on the rigid segment between the elbow and shoulder (Figure 3). This configuration allowed movement of the gripper (around the *z*-axes) to simulate lumbar rotation and movement of the elbow (around the *y*-axis) to simulate lumbar flexion. The SCORBOT ER VII was programmed to move to positions of approximately 15°, 30°, 45° and 60° of rotation around the *y*-axis and 0°, 15°, and 45° of rotation about the *z*-axis, displacing Vicon marker clusters and HTC Vive trackers synchronously.

### 2.5. Participants

Seven healthy adults between the age of 18 and 35 were recruited for this study. The exclusion criteria included (1) a history of low back injury that required medical attention, including chiropractic care or missing school or work, (2) current low back pain or low back pain in the last 6 months (only), or (3) any orthopedic or neurological impairment that would prevent participants from executing tasks that require moderate amounts of trunk flexion. The Institutional Review Board of Ohio University approved the study protocol and all participants signed an informed consent form. In compensation for their time, participants received an Ohio University Motor Control Lab t-shirt.

The participants were asked to wear shorts, a shirt and gym shoes provided by the laboratory. Vicon light-reflective trackers were attached to custom-designed 3D printed plates to create marker clusters for measuring 6-DOF. These plates were designed to allow the attachment of HTC Vive trackers in the center of the marker clusters, thus co-locating the two sensors. The 3D printed components were attached to the thorax and sacrum using elastic straps.

### 2.6. Procedure

(i) SCORBOT: With the sacrum and thorax components attached to the SCORBOT, four consecutive trials of 20 reaches were performed at 15°, 30°, 45° and 60° of rotation about the *y*-axis with five reaches for each 0°, 15°, 30°, 60° with additional rotation about the *z*-axis. For each reach, the motion was paused for 3 s at the target angle about the *y* and *x*-axis. Each movement started from a 0° flex/0° rotation position and data were recorded for the entirety of the movement.

(ii) Participants: One game of *Matchality* which consisted of two sets at each of the three levels of the game (See link to video in Appendix A). The location of the 4 × 4 set of blocks in this game was determined based on the participant’s arm length, trunk length, and hip height. Specifically, row 1 would be reached, in theory, with 15° of trunk flexion with the elbow extended and the shoulder flexed 90°, however, the body is a redundant system, leading to multiple degrees of freedom [27,35]. Rows 1–4 could be reached by flexing the trunk 15°, 30°, 45°, and 60° respectfully.

### 2.7. Outcomes

The time series data were examined to determine root means square (RMS) error and the mean difference at peak displacement between spatial position (position of the sacrum and thorax plates within the global coordinate system in cm) and orientation (rotation of the sacrum and thorax plates within the global coordinate system in degrees).

### 2.8. Data Collection

The time series position and orientation data for both systems were derived from the global coordinate system data with the Motion Monitor and Unity software. As the axes are different between the two systems, the data from the HTC Vive trackers were translated to conform to the world axes of the Vicon system before sending the data to the Motion Monitor software. The time-series Euler angle data and position data from the Vicon cluster and the HTC Vive trackers were exported from The Motion Monitor software. These time series data were imported into Matlab and processed using custom programs. All data were smoothed using a 40-point Savitzky–Golay [36] filter and DC offset removed. The two data sets were temporally aligned based on known events (initial start of game output from Unity game engine).

### 2.9. Statistical Analysis

RMS was computed on the time-series data from the Vive and Vicon positional and orientation data streams. The average RMS ± standard deviation (STD) is presented separately for the SCORBOT and for data collected on human participants. Separate 2-way repeated measures ANOVA (SPSS version 24.0, IBM) were used to assess differences in peak positional and orientation displacement between HTC Vive trackers and Vicon marker clusters for the SCORBOT and participants. Normality was assessed using standard skewness and kurtosis thresholds (i.e., <~1.25). The within-subject factors for the both the ANOVA for SCORBOT and the participants were System (i.e., Vicon, Vive) and target angle (15°, 30°, 45°, 60° target angles). Post hoc comparisons were assessed using Bonferroni test with adjustment for multiple comparisons. *p*-values < 0.05 were considered significant.

## 3. Results

Seven healthy adults were recruited for this study (see Table 2 for participant characteristics). The data of the SCORBOT is divided for position and orientation of the target angles 15°, 30°, 45° and 60°. Descriptive data for both experiments are reported as means ± standard deviation (see Table 3 and Table 4).

### 3.1. Position

Positional RMS error, averaged across thorax and sacrum components as well as across SCORBOT and participants was 0.58 ± 0.89 cm. Positional RMS error, averaged across SCORBOT and participants, is larger in thorax (0.77 ± 1.07 cm) component compared to the sacrum component (0.38 ± 0.61 cm). This is most likely driven by multi-planar movement at the thorax component compared to single plane motion of the sacrum component. Finally, RMS error was larger in participants game play (1.30 ± 0.92 cm) compared to the SCORBOT (0.02 ± 0.01 cm).

(i) SCORBOT: No main effect was found between the Vicon (0.46 ± 0.01 cm) and Vive (0.68 ± 0.24 cm) system on peak displacement of the thorax component (*p* = 0.37). Further, there was no interaction (*p* = 0.82) between the system and target angles (15°, 30°, 45°, 60° rotation about *y*-axis). However, as expected, there was a main effect of angle on peak displacement (F(3,6) = 2328.4, *p*< 0.001).

(ii) Participants: No main effect was found between the Vicon (0.24 ± 0.02 cm) and Vive (0.21 ± 0.04 cm) system at peak displacement of the thorax component (*p* = 0.41). While an interaction effect between the system and targets angles (15°, 30°, 45°, 60°) about *y*-axis trended toward significance (*p* = 0.053), pairwise comparison revealed no significant differences in peak displacement between Vicon and Vive for the four different target angles (e.g., 15°, p=0.36, 30°, p=0.42, 45°, p=0.42, 60°, *p* = 0.13).

### 3.2. Orientation

The average RMS error in orientation was 1.46 ± 0.59°. The difference between the two measurement methods was similar for participants (1.61 ± 0.62°) and the SCORBOT (1.66 ± 0.58°). The rotational RMS error was larger for the thorax component (1.90 ± 0.65°) compared to the sacrum component (1.38 ± 0.39°); however, as noted above, motion of the thorax component was multiplanar while the sacrum component was primarily about a single plane There was a noticeable increase of error in the SCORBOT when axial rotation was introduced (see Table 3 and Figure 4).

(i) SCORBOT: No main effect of system on the peak *y*-axis rotation of the thorax component (*p* = 0.073). On average, Vicon had a peak rotation of 33.2 ± 0.75° while Vive had peak rotation of 33.4 ± 0.79°. However, there was a significant interaction between the system and target angles (F (3,6) = 6.425, *p* = 0.027). A pairwise comparison showed the displacement measured between Vicon and Vive was significantly different only for 45°rotation about the *y*-axis (Vicon 41.88 ± 0.09°, Vive 42.12 ± 0.05°, F (1,8) = 8.703, *p* = 0.018). Finally, there was a significant main effect of target angle on peak rotation (F (3,6) = 389,022.8, *p* ≤ 0.001)

(ii) Participants: No main effect of system on peak *y*-axis rotation of the thorax component was found (*p* = 0.75) with the Vicon averaging 23.79 ± 3.58° and Vive 23.49 ± 4.10. Furthermore, there was no significant interaction between the system and target angles (*p* = 0.64). Finally, there was no main effect of target angle.

## 4. Discussion

This paper aimed to establish the agreement between a VR tracking system, the Vive HTC, with, a 3D optoelectric kinematic system (Vicon) during multiplanar dynamic tasks. As a traditional method of motion tracking, Vicon has established an accuracy of 0.1 mm and 0.1° within a calibrated volume space. This analysis shows that HTC Vive trackers agree with Vicon motion tracking with an average error of 0.58 ± 0.89 m and 1.46 ± 0.62° for position and orientation respectively. The inclusion of the SCORBOT robot allowed specifically controlled simultaneous rotations about two cardinal axes while natural human motion during gameplay in the VR space (i.e., while playing *Matchality*) provided comparison that would apply in more practical scenarios.

Positional accuracy of the HTC Vive trackers assessed using RMS of time series data ranged from 0.02 to 0.05 cm in the SCORBOT and 0.45 to 3.69 cm in participants, which is significantly larger compared to a study by Niehorster and colleagues [4] and Borges et al. [32]. They reported and RMS between 0.0049 cm and 0.0080 cm for the VIVE head-mounted display [31]. However, the large differences are most likely driven by key differences in methodology. Niehorster et al. used a median sample-to-sample RMS over three data points of 1s of stationary position of the HMD [31] compared to our use of 3 min of data during dynamic multiplanar movements. And Borges et al. used a custom tracking algorithm [32]. Besides the differences in analysis methods, these studies differed in the fact we used 3D dynamic motion and not just static and 2D motion [31,32].

Similar to the magnitude of position error, the average RMS error for rotation about the *y*-axis (i.e., flexion) measured between Vive and Vicon was 1.64 ± 0.62° which is larger than previously reported (0.0053 to 0.0111°) [31]. The larger RMS error values measured in the SCORBOT occurred with simultaneous rotation about the *z*-axis (i.e., axial rotation) and the *y*-axis (flexion). Specifically, as seen in Table 3, RMS error increased in SCORBOT_15 and SCORBOT_30 compared to SCORBOT_0. However, it is worth noting that RMS error was considerably smaller (i.e., 1.61 ± 0.62°) during human motions captured during VR gameplay compared to the RMS error of 2.56 ± 0.77° observed during the SCORBOR_30 condition.

Previous studies have tried to establish the accuracy and precision of position and orientation of the Vive HTC before [31,32,33]. However, these studies have reported various levels of positional and rotational accuracy (which will be discussed below) [31,33] and difficulty with reference planes when tracking of the HMD was lost [31]. We have not encountered this problem, possibly due to the (1) continuous updates provided by Steam VR solving initial orientation errors, (2) carefully setting up the play space, and (3) making sure the hand controllers are tracked well and are static during the setup process.

While differences between the Vive and Vicon exceeded previously described errors, visual examinations of the time series (Figure 4) suggest robust accuracy, especially in human participants engaged in typical motions (e.g., 15–45° of trunk flexion) in gameplay. Our results highlight the importance of considering movements about multiple axes simultaneously, as differences between the two systems increase. However, it is possible that these errors may be driven by differences in the method of calculating the Euler angle sequence between the systems (i.e., Vive sequence *x*, *y*, *z*, Vicon sequence *y*, *z*, *x*).

## 5. Conclusions

The HTC Vive trackers are wireless, lightweight, and inexpensive. They are easily attached to the body and allow reasonably accurate measures of joint motions (both position and orientation) during human movement with VR gameplay. While our results suggest that the accuracy of Vive trackers is lower than previously reported [31,32], they provide acceptable accuracy for tracking joint motion in human subjects using wireless body sensors (See Figure 4A). These data are very promising for future applications of this system for self-representation in the VR environment, clinical practice, and research labs. With the introduction of the second generation “Lighthouses”, additional “Lighthouses” can be linked covering up to a 10m diagonal space. This would increase tracking space and cater to experiments needing greater environments, multiple participants, or other clinical measures such as gait analyses. The ability to easily measure human motion using wireless sensors such as the Vive trackers, could have significant clinical implications and allow clinicians to quickly generate objective data to improve treatment. While these updates might not change the accuracy of positional and orientation data, they illustrate how fast VR equipment and software evolves. Accurate wireless tracking provides the ability to increase the agency and sense of presence in the VR environment, which has real-world implications for greater use of VR in rehabilitation as shaping of motor behavior within VR should then translate to improved movements in real life. As the development of new VR motion tracking devices is constantly evolving (e.g., finger tracking in new Valve Index controllers), the accuracy and fidelity of these devices will need to be assessed in controlled conditions. Future research should examine the role of multiple trackers accuracy tracking at higher movement velocities to further elucidate the potential of these motion tracking devices.

## Figures and Tables

**Figure 1 sensors-19-03632-f001:**
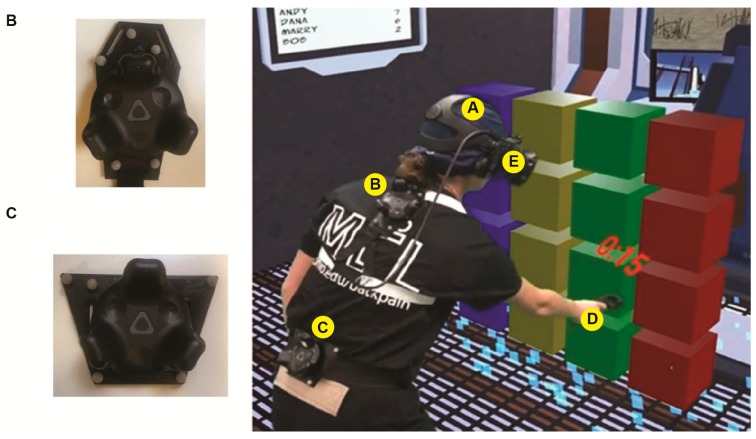
Participant instrumentation: (**A**) 3D printed plate for the head with Vicon markers (not compared in this in this study); (**B**,**C**) 3D printed plate with Vicon and Vive trackers for the thoracic and sacrum levels. (**D**) HTC Vive controller and (**E**) HTC Vive wired HMD.

**Figure 2 sensors-19-03632-f002:**
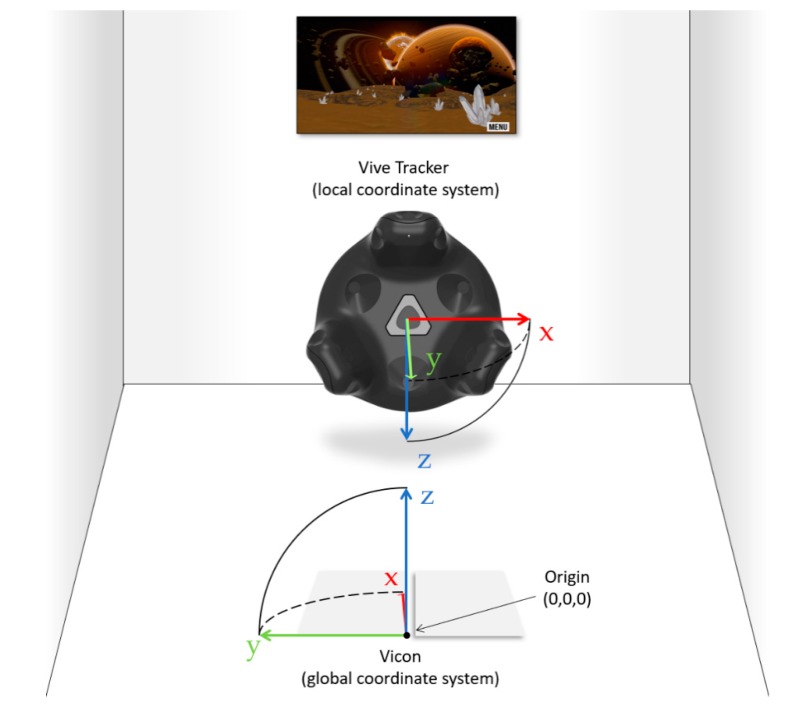
Illustration of Vive and Vicon coordinate systems.

**Figure 3 sensors-19-03632-f003:**
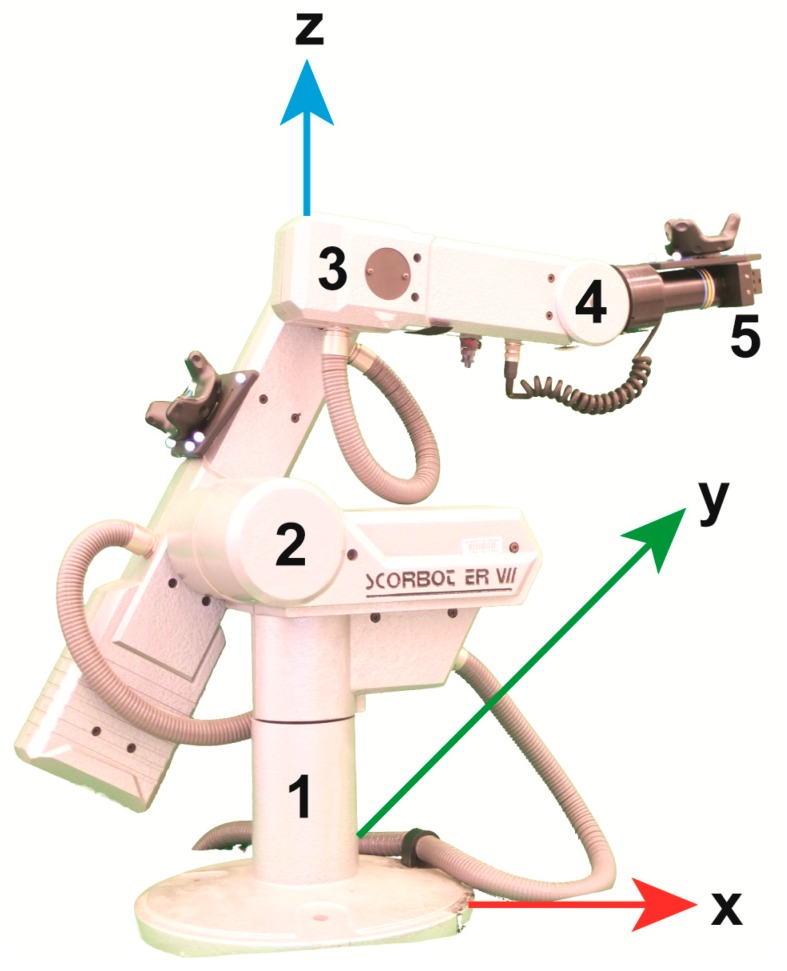
Representation of the SCORBOT ER VII. With Vicon axis system represented with the red (*X*), green (*Y*) and blue (*Z*) arrows. Sections of the scorebot: (1) Base—lower part of robot which rotates 310 about the *z*-axis, (2) Shoulder—connects to the base by way of a joint which rotates 35–130 about the *y*-axis, (3) Elbow—connects to the shoulder by way of a joint which also rotates 130 about the *y*-axis, (4) Wrist—connected to the elbow and gives the robot its final two degrees of freedom, rotating 360 about the *z*-axis and 130 about the *y*-axis, and (5) Gripper—the end effector attached to the wrist and capable of opening and closing.

**Figure 4 sensors-19-03632-f004:**
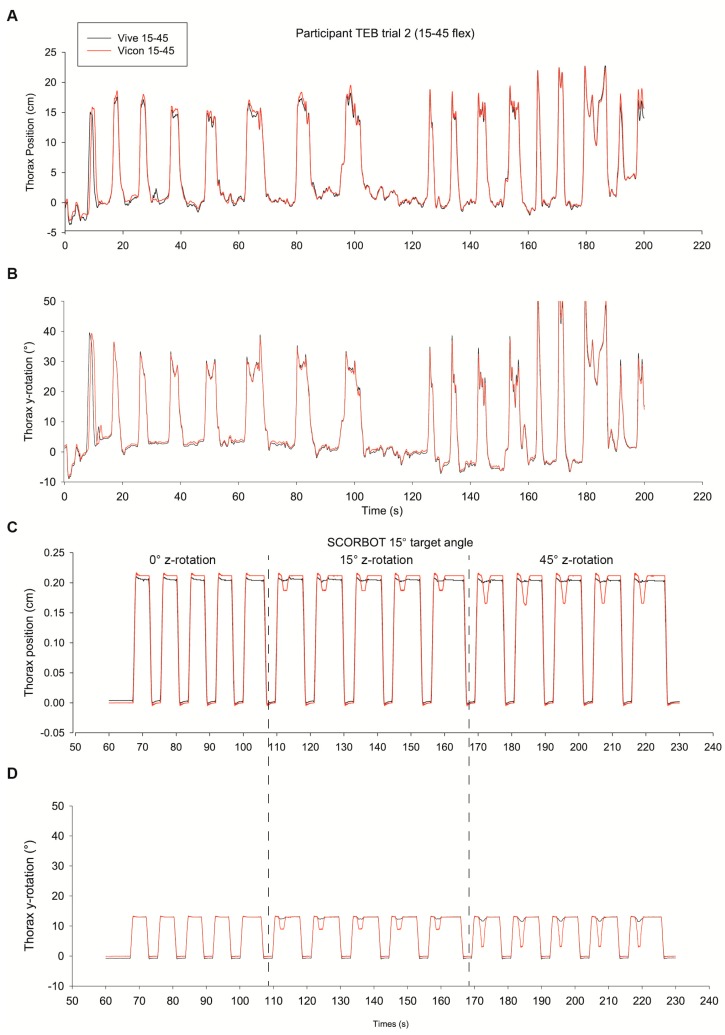
Time-series of Vive and Vicon. Graphs (**A**,**B**) show an example of thorax position and orientation of a participant playing *Matchality*. Graph (**C**,**D**) shows the thorax position and flexion angle of the SCORBOT moving 15° around the *y*-axis with 0°, 15°, and 45° rotation about the *z*-axis.

**Table 1 sensors-19-03632-t001:** Description of motion capture systems under study.

Characteristic	Vive	Vicon
Tracking system for lumbar motion	2 infrared laser lighthouses 2 Vive trackers (HTC America, Inc., Seattle, WA, USA)	10 Bonita 10 infrared cameras 10 light reflective markers mounted on 2 3D printed plates (Vicon Motion Systems Ltd., Oxford, UK)
Platform software	Steam VR (Valve Corporation, Washington, DC, USA).	Tracker version 3.4 (Vicon Motion Systems Ltd., Oxford, UK)
Motion capture software	Unity 2019.2.6f1 (Unity Technologies, California, CA, USA)	Motion Monitor (Innovative Sports Training, Chicago, IL, USA).
Sampling rate (Hz)	58–100	100
Latency (ms)	22	20
3D parameter	Euler angle (x, y, z)	Euler angle (y, z, x)

**Table 2 sensors-19-03632-t002:** Participant characteristics.

ID	Sex	Age	Weight (Kg)	Height (m)	BMI
1	F	24	59	1.60	23
2	F	23	81	1.68	29
3	M	25	82	1.75	27
4	M	22	77	1.89	22
5	F	41	90	1.68	32
6	F	22	57	1.68	20
7	M	30	59	1.70	20
**Mean**		72.14	72.14	1.71	24.71
**STD**		6.87	13.50	0.09	4.68

**Table 3 sensors-19-03632-t003:** Root mean square for Participants and SCORBOT with 0°, 15° and 45° of rotation about the *y*-axis of Vive and Vicon.

ID	Sacrum				Thorax			
	Position (mm)		Rotation (°)		Position (mm)		Rotation (°)	
	Mean	STD	Mean	STD	Mean	STD	Mean	STD
SCORBOT_0	0.02	0.00	1.65	0.52	0.02	0.02	1.21	0.34
SCORBOT_15	0.02	0.00	1.47	0.38	0.02	0.01	1.62	0.16
SCORBOT_30	0.02	0.01	1.47	0.37	0.03	0.02	2.56	0.77
Participants	0.67	0.69	1.18	0.33	1.74	0.96	1.98	0.54
average	0.18	0.61	1.44	0.39	0.45	1.07	1.84	0.65

**Table 4 sensors-19-03632-t004:** Average difference over 5 reaches between Vive and Vicon position and orientation of the thorax at peak displacement and orientation.

ID	15°		30°		45°		60°	
	Mean	STD	Mean	STD	Mean	STD	Mean	STD
	0.02	0.01	0.01	0.01	0.01	0.02	0.00	0.01
SCORBOT_0	0.02	0.01	0.01	0.01	0.01	0.02	0.01	0.01
SCORBOT_15	0.01	0.01	0.01	0.01	−2.81	4.89	0.03	0.05
SCORBOT_30	−0.01	0.02	−0.01	0.04	0.00	0.01	0.00	0.01
Participants								
average	0.01	0.02	0.00	0.03	-0.52	2.12	0.01	0.02
	**∆ (Vicon-Vive) Orientation**							
	0.38	0.59	−0.23	0.42	−0.37	0.40	−0.08	0.48
SCORBOT_0	0.21	0.43	−0.15	0.62	−0.24	0.19	−0.20	0.36
SCORBOT_15	−0.02	0.41	−0.43	0.28	−0.14	0.13	−0.35	0.31
SCORBOT_30	−0.11	0.87	1.42	5.95	2.16	4.64	−3.31	5.93
Participants								
average	0.06	0.66	0.06	0.66	0.14	4.68	0.81	3.19

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
