# Peer review of "Agreement Analysis between Vive and Vicon Systems to Monitor Lumbar Postural Changes"

_sensors, 2019, doi:10.3390/s19173632_

Round 1

Reviewer 1 Report

I am pleased to see that all comments were addressed properly. I thank you the Authors for their efforts.
I think this is a very improved version of the previously submitted review.
But I have further comments to add.
Page 5, Table 2: "72.14" is written in the cell of the mean of ages. It is wrong, please correct it!
Page 9, Table 2: I think the lines of this table are slipped, please check them.

Author Response

I am pleased to see that all comments were addressed properly. I thank you the Authors for their efforts.I think this is a very improved version of the previously submitted review.

RESPONSE:Thank you for your kind note

But I have further comments to add.

Page 5, Table 2: "72.14" is written in the cell of the mean of ages. It is wrong, please correct it!

RESPONSE: corrected this.

Page 9, Table 2: I think the lines of this table are slipped, please check them.

RESPONSE: corrected this.

Reviewer 2 Report

Dear authors,

I congratulate you for the great work done and I would like to suggest a few small changes to further improve your publication. I will follow the same revision format as the previous time:

Line 29 - Change "head mounted display" to "head-mounted display" (lines 50, 54,... too). Consider using (HMD) along the text: You defined HMD on line 49, but you don't use it... If it is repeated a lot you can use phrases like: vr device, immersive technology... (check 49-67)

Line 49 - HTC and Valve Corporation (two different companies developed this HMD) https://en.wikipedia.org/wiki/HTC_Vive

Lines 49 to 67 - You can simply use "Vive Tracker" to avoid write HTC repeatedly.

Line 68 - Please, provide a paragraph with a couple of examples of studies using Vicon. I need to be able to see that the Vicon is used in scientific experiments, even if it's obvious.

Line 124 - "30, 45º, and 60º" to "30º, 45º, and 60º". Or check consistency during the paper (same way whenever you list different degrees)

Figure 2- The text is not legible enough, consider moving it to the figure description or making it larger.

Line 176 - Remove this title subsection. The sections are too short and this can be joined.

Figure 4 - Can you re-scale the second graph to match the time scale? (A and B).

Line 217 - I think you need to develop this sentence a little further.

Line 233 - Subsection style or font is not consistent.

Line 318 - (Vive Vicon)?

Line 320 - Did you carried out the study with the Pro version? If so, make it clear in the introduction

Best regards.

Author Response

Dear authors,

I congratulate you for the great work done and I would like to suggest a few small changes to further improve your publication. I will follow the same revision format as the previous time:

Line 29 - Change "head mounted display" to "head-mounted display" (lines 50, 54,... too). Consider using (HMD) along the text: You defined HMD on line 49, but you don't use it... If it is repeated a lot you can use phrases like: vr device, immersive technology... (check 49-67)

RESPONSE: the head-mounted display is hyphenated now and HMD used throughout the manuscript to gain clarity.

Line 49 - HTC and Valve Corporation (two different companies developed this HMD) https://en.wikipedia.org/wiki/HTC_Vive

RESPONSE: We have changed this in the text.

Lines 49 to 67 - You can simply use "Vive Tracker" to avoid write HTC repeatedly.

RESPONSE: We have changed this in the text.

Line 68 - Please, provide a paragraph with a couple of examples of studies using Vicon. I need to be able to see that the Vicon is used in scientific experiments, even if it's obvious.

RESPONSE: We have provided a paragraph providing examples of the use of the vicon system and studies providing the accuracy optoelectic systems. The text starting at line 49 reads:

Motion capture setups are used in numerous fields. Studies based on motion capture data can be find in biomechanical, sport, or clinical science. Optoelectric motion capture based on markers was originally developed for gait analysis [25] and previously has been used to quantify lumbar flexion during reaching tasks [26–28]. Optical motion capture provides an powerful measuring opportunity for biomechanical applications accuracy [29] with an system error of less than 2mm [30].

Line 124 - "30, 45º, and 60º" to "30º, 45º, and 60º". Or check consistency during the paper (same way whenever you list different degrees)

RESPONSE: We have changed this in the text.

Figure 2- The text is not legible enough, consider moving it to the figure description or making it larger.

RESPONSE: We have changed this in the text.

Line 176 - Remove this title subsection. The sections are too short and this can be joined.

RESPONSE: We have changed this in the text.

Figure 4 - Can you re-scale the second graph to match the time scale? (A and B).

RESPONSE: We corrected the axes of graph B to match graph A.

Line 217 - I think you need to develop this sentence a little further.

RESPONSE: The sentence starting with Participant characteristics are presented…. Is removed from the statistical section. Under the results section a reference is made to the participant characteristics and data tables.

Seven healthy adults were recruited for this study (see Table 2 for participant characteristics). The data of the SCORBOT is divided for position and orientation of the target angles 15º, 30º, 45º and 60º. Both, the data of the participants and SCORBOT, are presented as mean, standard deviation (STD) in table 3 and 4.

Line 233 - Subsection style or font is not consistent.

RESPONSE: This is corrected in the text

Line 318 - (Vive Vicon)?

RESPONSE: both systems use a different Euler angle sequence which we have specified in the text now: (Vive sequence x, y, z, Vicon sequence y, z, x)

Line 320 - Did you carried out the study with the Pro version? If so, make it clear in the introduction

RESPONSE: We used the original HTC Vive headset and pucks, however the same pucks are developed to pair with the Vive Pro.

Best regards.

This manuscript is a resubmission of an earlier submission. The following is a list of the peer review reports and author responses from that submission.

Round 1

Reviewer 1 Report

The article is about comparing the Vive tracker to the Vicon system. 

In spite of the fact that the paper is very interesting and it could be very necessary paper, the current state of the manuscript has some deficiencies.

The introduction is not enough wide range of literature review. I can recommend your e.g. the ICDVRAT conference proceedings. You can find a lot of VR/AR healthcare related publication in the ICDVRAT conference series archive proceedings: https://www.icdvrat.org/archive.htm 

It could be better if you insert the table about the data of participants not in the Appendix but after the 109 line on the 4th page. 

Anyway, table 1 in the Appendix is not correct. 

Page 11, Table A1: The data of the Mean of the Age and the Mean of the Weight are not correct. Please check them.

Page 7, line 227: What do you mean about Human subjects? Do you mean only the positions or also other data? Please, explain a bit more detailed.

Page 9, line 243: Why do you say "golden standard" about the Vicon system?

To sum it up, after correcting the above-mentioned mistakes, I can suggest the paper for publication.

Reviewer 2 Report

Dear authors,

You have presented a very interesting study that compares two tracking systems when monitoring lumbar postural movements. The study seems well designed and presented, using two different experiments for the data analysis. In one of them, a videogame has been developed to gamify the postural analysis. In the other, a robotic arm has been used to simulate the different positions. In addition they show a video in which it is possible to appreciate the true complexity of the development carried out. The study of VR devices is a sector of current relevance, so the article is of scientific interest.

The article is well presented and easy to read, although I believe that substantial improvements should be made to its publication. These are the changes I consider necessary to make this article a remarkable work.

Introduction:

line 28 - Please, order the simulations, the vision is the first sense that simulates the VR. Examples and references are needed. You can talk about stereoscopy and binaural sound in real time.

line 29 - "it is now expanding" to "now it is expanding". Sounds better that way.

line 36 - “clinicians and researcher” ending with "s."

line 37 - This HMD was released by HTC and Valve, forming HTC Vive.

line 38 - "less than a thousand US dollars" interesting way to compare it with Vicon, but I think it's not formal enough. It is clearer if you indicate that it costs less than the Vicon directly.

Line 41 - Develop reference 6 further. It is interesting to note that the HMD has tracking problems.

General: The device is called Vive tracker. Try to use this term where it makes sense, so as not to be so repetitive with: "HTC Vive tracker punk".

General: I need to develop the state of the art a little more. You can talk about other studies of positional tracking systems, other studies done with the Vicon or similar devices, how they are measured and evaluated. It's very interesting what you say, but I see a lack of context.

Materials and Methods:

line 62 - Write an introductory paragraph that serves as a thread with the following.

line 66 - Talk about HTC Vive Lighthouses, since it is the name of this technology.

line 72 - Reference is needed to affirm: "This architectural change does not alter the punk spatial or temporal accuracy".

line 79 - The phrase is not well constructed.

Line 82 - Please detail the table further, sections such as "Connections" are not well referred to.

Figure 1- Detail the Figure. The coordinate system of the Vive Tracker can be shown in a single image, just like the Vicon. It would make it easier to compare.

line 98 - "30,45, and 60º" to "30º,45º, and 60º" (check consistency along the paper).

line 99 - How do you ensure in the methodology that the user does not bend over to avoid lumbar flexion?

line 111 – What is Vive Bonita?

line 112 - "providing an adequate data collection", please provide a reference: what is adequate?

line 130 - "move to positions of approximately". Please, provide more information and references.

Figure 2-Move Figure text inside the description.

line 136 and 137- Please, use (1), (2) or (i), (ii)... It's weird to find an unencapsulated number. The same in the abstract section.

line 137 - Point 2: low back pain in the last 6 months (only).

line 142 - I'm sorry, but I don't think it's relevant. I find it more interesting that you add a more detailed description of the mean data and user deviations at this point, removing table A1 from the appendix section.

line 142-Same section as "participants".

line 150-Procedure introduction paragraph, later use “latex itemize” to separate.

line 172 - 40-point Svitzky-Golay: Can this smoothing affect the results? How is it done in other studies? I need references to check that the methodology is correct.

line 175 - References, I need a more detailed explanation. State of the art don’t support supplied methodology.

Line 185 - Is it a parametric sample? Has any test been carried out to check if the application of ANOVA is adequate?

Results:

Line 214 - Avoid using personal appraisals or use a reference. "orientation was moderate" What do you base it on?

line 220 and 227-"Latex itemize"

Table 2 and 3: I don't think it's necessary to show all the data. Raise the table showing averages and deviations. This can be in appendices or not even that... Both tables could be a single table showing less data. Improve descriptions and fix typos: 'ID.

Figure 4 - Please, improve the composition, that the axes have the same scale to be able to be compared (A and B).

Conclusions:

Please provide more conclusions or future lines of research.

I think Appendix A1 is not necessary.

General: A small revision of grammar is needed, as there are some errors. Check references (i.e. ref 9...) and white spaces when referencing them within the text (i.e., line 253, line 256, line 258, line 31...).